# Uncovering Implicit Bias in LLM Mathematical Reasoning with Concept Learning

## Abstract

We introduce a new framework of concept learning tasks that helps uncover implicit biases in large language models. Using in-context concept learning experiments, we found that language models can have a bias toward upward monotonicity; such bias is less apparent when the model is tested by direct prompting without concept learning components. This demonstrates that in-context concept learning can be an effective way to discover hidden biases in large language models.

## 1 Introduction

As Large Language Models (LLMs) become important components of various Natural Language Processing (NLP) systems, there is an increased scrutiny on the potential biases models may have before deployment. Researchers have developed benchmarks and de-biasing methods that help to detect and mitigate biases in LLMs [Wan et al., 2023, Hofmann et al., 2024, Shirafuji et al., 2025]; however, recent studies show that models that appear unbiased on standard benchmarks can still have implicit biases that are hard to detect [Bai et al., 2025, Tan and Lee, 2025].

To help uncover such implicit biases, we propose a new method inspired by the literature on human concept learning [Feldman, 2000, Goodman et al., 2015, Piantadosi et al., 2016]. In this paper, we use in-context concept learning to show how LLMs can have an implicit bias toward certain mathematical concepts.

Consider the prompt in (1). In the first four lines, we see two labeled examples of an unknown mathematical concept, expressed by "the desired quantity". The last two lines ask the model to label a new example. By repeating this process for a variety of scenarios and concepts, we can study whether the model achieves better accuracy with mathematical concepts that have certain properties.

(1)  There are 10 boxes. Alice has 5 of the 10 boxes. Does Alice have the *desired quantity* of the boxes? **No**.
There are 15 boxes. Alice has 8 of the 15 boxes. Does Alice have the *desired quantity* of the boxes? **Yes**.
There are 16 boxes. Alice has 9 of the 16 boxes. Does Alice have the *desired quantity* of the boxes? ___

We will use the idea of monotonicity from the study of semantics to formulate our experiments. Here, we give an informal introduction to semantic monotonicity in quantifiers, which are semantic objects that describe the relation between two sets of objects in a discourse [Barwise and Cooper, 1981]. For example, *more than half* is a quantifier in English, which can be a possible meaning for the unknown concept in (1), since we can substitute *more than half* for *the desired quantity* in the first two examples while satisfying their corresponding labels. A quantifier is upward monotone if

inferences from subsets to supersets are valid [1], and it is downward monotone if inferences from supersets to subsets are valid [Icard III and Moss, 2014, Carcassi et al., 2021]. See table 1 for some concrete examples.

| Upward monotone | |
|---|---|
| more than $n$ | More than 5 boxes are in Berlin. $\Rightarrow$ More than 5 boxes are in Germany. |
| some | Some boxes are in Berlin. $\Rightarrow$ Some boxes are in Germany. |
| *Downward monotone* | |
| less than $n$ | Less than 5 boxes are in Germany. $\Rightarrow$ Less than 5 boxes are in Berlin. |
| no | No boxes are in Germany. $\Rightarrow$ No boxes are in Berlin. |

Table 1: Monotonicity in quantifiers.

We test three different LLMs on a range of mathematical concepts that differ in semantic monotonicity, and find that models tend to have greater success with upward monotone concepts. We then performed the same set of experiments with explicit semantics, that is, instead of hiding the meaning of the concept using the phrase "the desired quantity", we replaced the phrase with the description of the concept's meaning in plain English[2]. We find that the bias toward upward monotonicity becomes less noticeable. This suggests that concept learning can be used to uncover implicit biases in mathematical reasoning that are hard to detect with standard methods.

## 2   Related Work

Min et al. [2022], Wei et al. [2022], Kojima et al. [2022], among many others, have studied how and why in-context learning in LLMs works. Bai et al. [2025] demonstrated that LLMs that appear unbiased on common social bias (such as gender and race) benchmarks can still have implicit biases in their decision-making process. Our work follows a similar theme, but instead of social biases, we study implicit biases in LLMs' mathematical reasoning process. Geurts and van der Slik [2005] showed that humans generally achieve higher accuracy for upward monotone quantifiers (vs. downward monotone) in quantifier inference tasks, which suggests downward monotone quantifiers can be more cognitively complex for humans. Jumelet et al. [2021] studied how LLMs use semantic monotonicity to process negative polarity items in English. And Wang et al. [2024] showed that similar to humans, LLMs can have a simplicity bias in concept learning that favors logically simpler concepts.

## 3   Methodology

### 3.1   Concept selection

For concepts that are upward monotone, we use the quantifier "more than $p$", where $p$ represents a proportion of the total number of items. For downward monotone concepts, we use the quantifier "less than $p$". We use the following values for $p$ in all experiments:

p = {1/10, 2/10, 3/10, 4/10, 5/10, 6/10, 7/10, 8/10, 9/10}.

See Appendix B for additional proportional concepts and cardinal concepts (e.g. "more/less than $c$" where $c$ is a cardinal number) results.

### 3.2   Prompt generation

In each prompt, we include 20 labeled examples (10 with positive labels and 10 with negative labels) in a random order. Each example is generated using the template in (2), where the underlined slots denote the linguistic items that may vary between examples. See C in Appendix for an example prompt used in the experiments.

---

[1]In this paper, we adopt a simplified definition of quantifier monotonicity, and only consider the monotonicity of the second argument taken by the quantifier. For a more rigorous treatment, see Icard III and Moss [2014].

[2]e.g. the question in the prompt can be "Does Alice have *more than 1/2* of the boxes?"

(2)  There are ___________ ___________. Alice has ___________ of the ___________ ___________.
TOTAL     OBJ                                                    NUM              TOTAL        OBJ

Does Alice have the desired quantity of the _________? _________.
OBJ                    YES/NO

We iterate through all meaningful[3] numerical ranges for both the number of total objects, and the number of objects Alice has, and generate an example for each combination. For each prompt, we randomly sample without replacement 10 positive examples and 10 negative examples from the sets of all possible positive examples and negative examples respectively. An unseen example is used as the question at the end.

For the set of objects in the prompt, we first took all the countable nouns from The Oxford 3000 word list, which is a (public-domain) list of 3000 most important words to learn in English, chosen by language experts at Oxford Dictionaries. [4] We then ranked the nouns by their unigram frequency in OLMo-2-32B's training corpus, and chose the top 100 most frequent nouns as the set of objects in the prompt. For each example, the object is randomly sampled from the set of 100 nouns.

### 3.3  Evaluation

To determine the model's response to a prompt, we compare the probabilities (computed by the model) of the prompt with a positive response appended at the end, with the same prompt with a negative response appended. In other words, if $P(prompt + \text{``Yes''}) > P(prompt + \text{``No''})$, then we consider the model has given a positive response to the prompt. For each concept, there are 250 prompts with positive true labels and 250 prompts with negative true labels. As a baseline (random), a model that answers every prompt by flipping a fair coin would achieve $50\%$ accuracy.

### 3.4  Models

We ran experiments on two LLM families that are fully open-source: OLMo 2 [OLMo et al., 2024] from Allen Institute for AI (Ai2) and K2 [Liu et al., 2025] from LLM360. OLMo 2 is a family of fully open language models that achieve competitive performance on various benchmarks when compared to other open-weight models. K2-65B is a fully open model that outperforms Llama-2-70B [Touvron et al., 2023] on most benchmarks while using less compute. We chose models that are fully open (open model weights, model checkpoints, and exact training data) so that our experiments will have better reproducibility; having full access to the model training data will also make explaining model behaviors from training data possible in future work.

## 4  Results

For each model, we run two sets of experiments – one with concept learning and the other with explicit semantics. In figure 1, the two charts in the first vertical column show the results for OLMo-2-13B-instruct. We see that the accuracy differences between concepts [more than $p$] and [less than $p$] are much greater in the concept learning set (top chart) when compared to the explicit semantics set (bottom chart). Similar patterns can be observed for OLMo-2-32B-instruct, represented by the two charts in the second vertical column. In the third column, we see that K2-65B[5] also has a bias toward upward monotone concepts, although the difference in accuracy between the two classes of concepts is smaller when compared to OLMo-2. This suggests that K2-65B may have a weaker bias toward upward monotonicity.

---

[3]e.g. Alice cannot have more items than the total number of items, or have less than 0 items. We also require the total number of items to be between 5 and 100 inclusive.

[4]`https://www.oxfordlearnersdictionaries.com/us/wordlist/american_english/oxford3000/`

[5]checkpoint 300 (out of 380) was used in this experiment

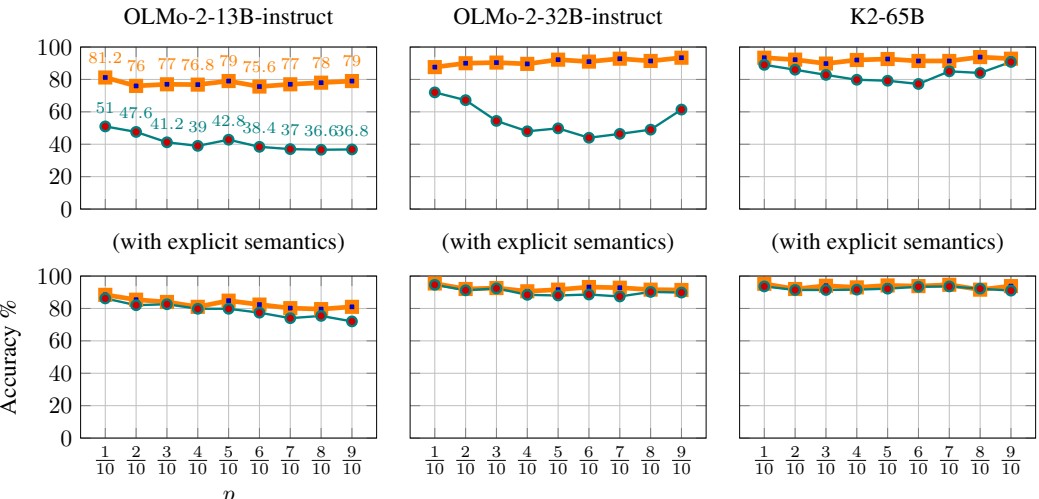

Figure 1: In-context concept learning helps uncover bias in monotonicity that is less noticeable in standard evaluation methods. In experiments with OLMo-2 and K2, models tend to have higher accuracies with upward monotone concepts during concept learning experiments (top). However, this bias is less noticeable in explicit semantics experiments (bottom).

## 5 Discussion

The results with OLMo-2 and K2 models show that some LLMs consistently achieve lower accuracies for downward monotone quantifiers when learning new concepts. The precise reason for this phenomenon is not yet known, but we have developed the following hypothesis:

- [Agmon et al., 2019, 2022] have shown that downward monotone quantifiers can be expressed as the *negation* of their upward monotone counterparts. They showed that humans generally require more processing time for downward monotone quantifiers (compared to upward monotone ones) in quantifier verification tasks, and further hypothesized that the hidden negation operation contributed to the increased complexity of downward monotone quantifiers.
- Wang et al. [2024] showed that LLMs can be biased toward logically simpler concepts when performing concept learning tasks.

Based on the two observations above, we hypothesize that downward monotone concepts can be considered more logically complex than upward monotone ones in LLM concept learning because of the hidden negation operator. And similar to humans, certain LLMs perform worse on downward monotone concepts since they are biased toward logically simpler concepts, such as their upward monotone counterparts.

To summarize, we make the following contributions in this paper:

- We show that LLMs can have an implicit bias toward mathematical concepts that are upward monotone.
- We demonstrate that concept learning can be a new tool to uncover such bias.
- Lastly, we compare LLMs' performance with human performance on similar mathematical concepts, and find that LLMs have human-like biases in mathematical concept learning.

We hope this work will inspire new inquiries into uncovering hidden biases and studying how training data can enable certain biased behaviors in large language models.

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

## A Limitations

Only a limited set of LLMs were tested; we hope to apply the concept learning methodology to study a larger set of models with greater variety as future work.

The set of upward/downward monotone concepts is relatively small. The next step can be to study this phenomenon with more complex concepts with varying monotonicity.

## B Additional concept learning results

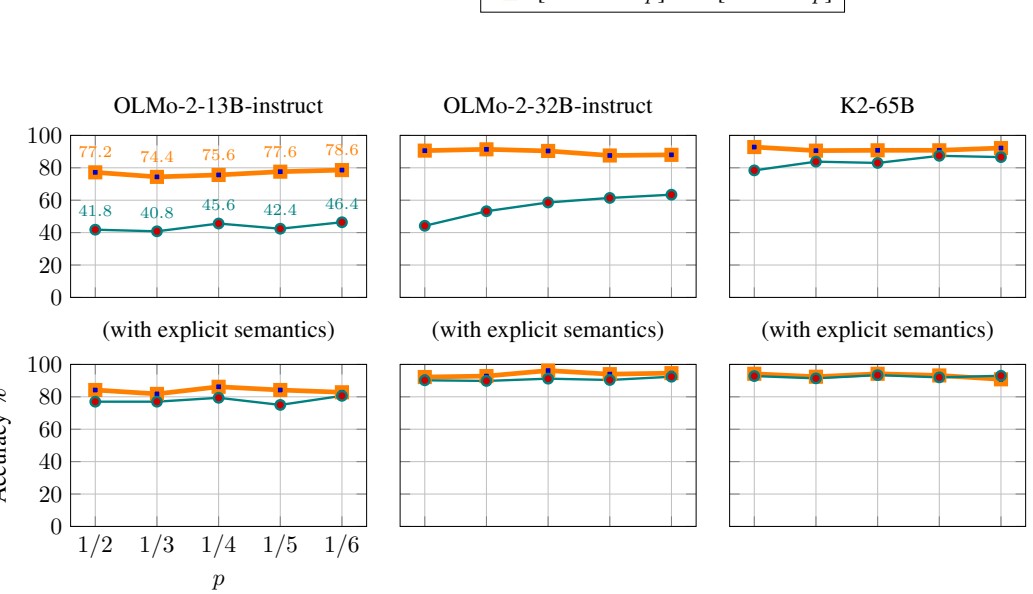

Figure 2: In-context concept learning helps uncover bias in monotonicity that is less noticeable in standard evaluation methods. In experiments with OLMo-2 and K2, models tend to have higher accuracies with upward monotone concepts during concept learning experiments (top). However, this bias is less noticeable in explicit semantics experiments (bottom).

Table 2: Accuracy values

| Model | 1/2 | 1/3 | 1/4 | 1/5 | 1/6 |
|---|---|---|---|---|---|
| **OLMo-2-13B-instruct** | | | | | |
| [more than p] | 77.2 | 74.4 | 75.6 | 77.6 | 78.6 |
| [less than p] | 41.8 | 40.8 | 45.6 | 42.4 | 46.4 |
| **OLMo-2-13B-instruct (explicit semantics)** | | | | | |
| [more than p] | 84.2 | 81.8 | 86.2 | 84.2 | 82.8 |
| [less than p] | 77.0 | 77.4 | 79.4 | 75.0 | 80.6 |
| **OLMo-2-32B-instruct** | | | | | |
| [more than p] | 90.6 | 91.4 | 90.4 | 87.6 | 88.0 |
| [less than p] | 44.2 | 53.2 | 58.6 | 61.4 | 63.4 |
| **OLMo-2-32B-instruct (explicit semantics)** | | | | | |
| [more than p] | 92.2 | 92.8 | 96.2 | 94.0 | 94.6 |
| [less than p] | 90.2 | 89.8 | 91.2 | 90.4 | 92.4 |
| **K2-65B** | | | | | |
| [more than p] | 92.8 | 90.6 | 90.8 | 90.8 | 92.2 |
| [less than p] | 78.4 | 83.8 | 83.0 | 87.4 | 86.6 |
| **K2-65B (explicit semantics)** | | | | | |
| [more than p] | 94.2 | 92.4 | 94.2 | 93.2 | 90.8 |
| [less than p] | 92.8 | 91.4 | 93.4 | 92.0 | 93.0 |

### B.1 Cardinal concepts

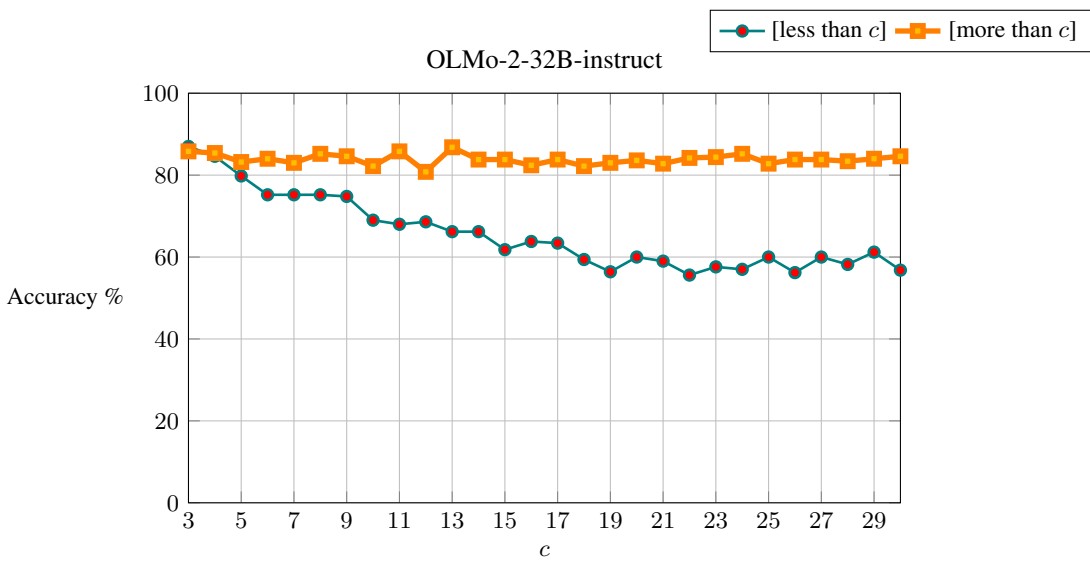

Figure 3: OLMo-2-32B concept learning results with cardinal concepts. The bias toward upward monotonicity becomes stronger as $c$ increases. $c \in [3, 30]$.

 ## C Example prompt

| Prompt | Label |
|---|---|
| There are 76 fields. Alice has 65 of the 76 fields.
Does Alice have the desired quantity of the fields? No.
There are 74 houses. Alice has 17 of the 74 houses.
Does Alice have the desired quantity of the houses? Yes.
There are 51 rolls. Alice has 18 of the 51 rolls.
Does Alice have the desired quantity of the rolls? Yes.
There are 79 beds. Alice has 52 of the 79 beds.
Does Alice have the desired quantity of the beds? No.
There are 89 boards. Alice has 46 of the 89 boards.
Does Alice have the desired quantity of the boards? No.
There are 82 books. Alice has 42 of the 82 books.
Does Alice have the desired quantity of the books? No.
There are 96 texts. Alice has 6 of the 96 texts.
Does Alice have the desired quantity of the texts? Yes.
There are 94 cars. Alice has 49 of the 94 cars.
Does Alice have the desired quantity of the cars? No.
There are 81 forces. Alice has 65 of the 81 forces.
Does Alice have the desired quantity of the forces? No.
There are 81 whiles. Alice has 75 of the 81 whiles.
Does Alice have the desired quantity of the whiles? No.
There are 90 endings. Alice has 52 of the 90 endings.
Does Alice have the desired quantity of the endings? No.
There are 79 lips. Alice has 3 of the 79 lips.
Does Alice have the desired quantity of the lips? Yes.
There are 17 men. Alice has 4 of the 17 men.
Does Alice have the desired quantity of the men? Yes.
There are 42 states. Alice has 36 of the 42 states.
Does Alice have the desired quantity of the states? No.
There are 73 rooms. Alice has 29 of the 73 rooms.
Does Alice have the desired quantity of the rooms? Yes.
There are 64 waters. Alice has 11 of the 64 waters.
Does Alice have the desired quantity of the waters? Yes.
There are 50 friends. Alice has 8 of the 50 friends.
Does Alice have the desired quantity of the friends? Yes.
There are 28 rooms. Alice has 8 of the 28 rooms.
Does Alice have the desired quantity of the rooms? Yes.
There are 56 bits. Alice has 56 of the 56 bits.
Does Alice have the desired quantity of the bits? No.
There are 96 apps. Alice has 9 of the 96 apps.
Does Alice have the desired quantity of the apps? Yes.
There are 95 boxes. Alice has 7 of the 95 boxes.
Does Alice have the desired quantity of the boxes? | |
| | Yes |

Table 3: An example of prompts in the dataset. The concept in this example is "less than half". In an *explicit semantics* experiment, the prompt will have a similar structure, but the phrase "the desired quantity" will be replaced by "less than 1/2".

