# OpenReview forum: "Uncovering Implicit Bias in LLM Mathematical Reasoning with Concept Learning"
_EurIPS.cc/2025/Workshop/UPLB — UPLB2025_

### Official Review · Reviewer_BrQA · 2025-11-03
**Promising hypothesis on implicit monotonicity bias in LLMs, but evidence is very preliminary**

**Rating:** 5
**Confidence:** 3

**Review:**

This paper investigates whether large language models exhibit an implicit preference for upward monotonic concepts such as “more than half” compared to downward monotonic ones like “less than half” when learning from examples. The authors create small synthetic datasets of fraction-comparison tasks, where models must infer these concepts from labeled examples. They evaluate OLMo-2-13B, OLMo-2-32B, and K2-65B in few-shot settings (20 demonstrations per prompt), measuring accuracy through the relative log-probabilities of “Yes” versus “No” completions. The models perform better on upward monotonic concepts, suggesting an internal bias toward upward monotonic reasoning that vanishes when the rule is explicitly described in the prompt.

I find the idea interesting and clear, but the experiments are limited in scope and control. The paper lacks ablation analyses to rule out simpler explanations, for example, that models might learn superficial cues from prompt wording or label biases rather than genuine monotonicity reasoning. Overall, the paper raises an interesting hypothesis, motivated by similar asymmetries observed in human reasoning, and is relevant to the workshop's theme of understanding inductive biases in LLMs. However, the scope feels more exploratory than conclusive, stronger controls and broader validation would be needed to support the conclusions.

---

### Decision · Program_Chairs · 2025-11-03

Accept (Poster)